# VCP inhibition induces an unfolded protein response and apoptosis in human acute myeloid leukemia cells

Paweł P. Szczęśniak[1], Jan B. Heidelberger[2], Hubert Serve[1,3,4], Petra Beli[2,5], Sebastian A. Wagner [1,3,4]*

1 Department of Medicine, Hematology/Oncology, Goethe University, Frankfurt, Germany, 2 Institute of Molecular Biology (IMB), Mainz, Germany, 3 German Cancer Consortium (DKTK) and German Cancer Research Center (DKFZ), Heidelberg, Germany, 4 Frankfurt Cancer Institute (FCI), Frankfurt, Germany, 5 Institute of Developmental Biology and Neurobiology (IDN), Johannes Gutenberg University Mainz, Mainz, Germany

* swagner@med.uni-frankfurt.de

## Abstract

Acute myeloid leukemia (AML) is a heterogeneous malignancy characterized by the accumulation of undifferentiated white blood cells (blasts) in the bone marrow. Valosin-containing protein (VCP) is an abundant molecular chaperone that extracts ubiquitylated substrates from protein complexes and cellular compartments prior to their degradation by the proteasome. We found that treatment of AML cell lines with the VCP inhibitor CB-5083 leads to an accumulation of ubiquitylated proteins, activation of unfolded protein response (UPR) and apoptosis. Using quantitative mass spectrometry-based proteomics we assessed the effects of VCP inhibition on the cellular ubiquitin-modified proteome. We could further show that CB-5083 decreases the survival of the AML cell lines THP-1 and MV4-11 in a concentration-dependent manner, and acts synergistically with the antimetabolite cytarabine and the BH3-mimetic venetoclax. Finally, we showed that prolonged treatment of AML cells with CB-5083 leads to development of resistance mediated by mutations in *VCP*. Taken together, inhibition of VCP leads to a lethal unfolded protein response in AML cells and might be a relevant therapeutic strategy for treatment of AML, particularly when combined with other drugs. The toxicity and development of resistance possibly limit the utility of VCP inhibitors and have to be further explored in animal models and clinical trials.

## Introduction

The ubiquitin-proteasome system (UPS) is a non-lysosomal proteolytic cascade in which ubiquitin E3 ligases transfer ubiquitin onto substrate proteins and thereby target them for degradation [1]. Cancer cells are characterized by uncontrolled growth and frequently rely on rapid protein turnover. Targeting the UPS has emerged as a therapeutic strategy for cancer: the proteasome inhibitor bortezomib is approved for treatment of multiple myeloma (MM) and

**Data Availability Statement:** All relevant data are within the paper and its Supporting information files.

**Funding:** The study was supported by the ProLOEWE Ubiquitin Networks (Ub-Net) research cluster of the State of Hessen and by a project grant from the Else Kröner-Fresenius Foundation awarded to SW (2015_A124). The funders had no role in study design, data collection and analysis, decision to publish, or preparation of the manuscript.

**Competing interests:** The authors have declared that no competing interests exist.

mantle cell lymphoma (MCL). More recently, additional proteasome inhibitors including carfilzomib and ixazomib have been developed and approved for clinical use in MM [2–4].

Valosin-containing protein (VCP) (also known as p97 or CDC48 in yeast) is a key component of the UPS that cooperates with a group of adaptor proteins to extract substrates from multi-subunit protein complexes and cellular compartments, unfolds them, and subjects them to degradation by the 26S proteasome. VCP ensures protein quality control within the endoplasmic reticulum (ER) by extracting misfolded proteins from the ER membrane in a process termed endoplasmic reticulum-associated degradation (ERAD). VCP is also involved in the regulation of autophagy, receptor-mediated endocytosis, and chromatin-associated degradation of proteins involved in DNA replication and DNA damage repair [5, 6].

VCP belongs to the adenosine triphosphates (ATPases)-associated with various cellular activities (AAA+) family and is composed of two ATPase domains (D1 and D2) as well as the N-domain [7]. VCP inhibition can be achieved by a range of small molecule inhibitors including NMS-873 (allosteric, non-competitive mode of action) [8] and CB-5083 (reversible ATP competitive mode of action), the latter being an orally-bioavailable compound developed by Cleave Biosciences [9, 10]. Inhibition of VCP has been tested in diverse cancer cell lines and tumor models: CB-5083 has been suggested to decrease cell growth in MM [11], acute lymphoblastic leukemia (ALL) [12], and lymphoma [13, 14]. A recent study suggests that inhibition of VCP disrupts the cellular DNA repair and thereby inhibits proliferation of AML cells [15].

Acute myeloid leukemia (AML) is a heterogeneous malignancy characterized by the accumulation of undifferentiated white blood cells (blasts) in the bone marrow. In adults, who develop AML at the age of 60 or more, current treatment strategies are able to induce remissions, however frequent relapses lead to 2-year survival rate of only 10%-50% [16]. Historically AML has been treated with combination chemotherapy consisting of the antimetabolite cytosine β-D-arabinofuranoside (cytarabine) and the anthracycline daunorubicin. More recently, with the advent of high-throughput tumor genome sequencing, the genetic heterogeneity of AML has been explored. Re-occurring genetic alterations identified in AML are mutations in components of the signal transduction (e.g. FLT3, KIT, KRAS, NRAS), nulceophosmin-1 (NPM1), myeloid transcription factors, chromatin modifiers, tumor suppressors (e.g. TP53, WT1) and genes associated with DNA methylation (e.g. TET1, TET2, IDH1, IDH2, DNMT3B, DNMT1, DNMT3A) as well as gene fusions involving transcription factors. The identification of reoccurring genetic alterations has spurred the development of targeted therapies for AML and by now multiple drugs have entered the clinic: the tyrosine kinase inhibitors midostaurin and gilteritinib have proven efficacious in the treatment of patients harboring FLT3 mutations and mutation-specific inhibitors of IDH1 and IDH2 have shown promising activity in patients with relapsed AML. The BH3-mimetic venetoclax that blocks the anti-apoptotic B-cell lymphoma-2 (Bcl-2) protein and has recently shown great promise in the treatment of patients with relapsed AML including previously difficult to treat subgroups with TP53 alterations [10, 17].

There have been also attempts to target components of the ubiquitin-proteasome system in AML: the MDM2 (E3 ligase) antagonist RG7112 exerts antileukemic effects by stabilizing p53 (Roche, phase 1) [18]. Pevonedistat (TAK-924 or MLN4924) inhibits cullin-RING E3 ligases by destabilizing neural precursor cell expressed developmentally down-regulated protein 8 (NEDD8)-activating enzyme (NAE), leading to ubiquitin activation inhibition and decreased AML growth (Takeda, currently in phase 1, 2 and 3) [19]. Recently, inhibitors of VCP have shown efficacy in preclinical AML models and a phase I trial with the VCP inhibitor CB-5083 has been performed (NCT02243917 and NCT02223598). However, the clinical development of CB-5083 has been stopped due to off-target effects resulting in vision problems. An improved molecule developed by the same company recently entered phase I for the treatment of AML, MDS and solid cancers (NCT04402541 and NCT04372641) [15].

In a targeted small-hairpin RNA (shRNA)-based genetic screen for UPS-related genes required for AML cell proliferation and survival we identified *VCP* as a candidate gene. In this work we demonstrated that *VCP* inhibition using the *VCP* inhibitors CB-5083 and NMS-873 leads to accumulation of ubiquitylated proteins, activation of the unfolded protein response and induction of apoptosis in AML cells. We also showed a synergistic effect of VCP inhibition with cytarabine and venetoclax that are currently used in the treatment of AML. Finally, we established CB-5083 resistant cell lines to the investigate mechanism that can lead to development of resistance against VCP inhibition. In summary, our results provide further insights into the role of VCP inhibition in AML and show that mutations in *VCP* can provide a resistance mechanism to VCP inhibitors in AML cells.

## Results

### VCP inhibition induces an unfolded protein response in AML cells

To assess the short-term effects of VCP inhibition, we treated acute myeloid leukemia (AML) cells with the VCP inhibitors NMS-873 and CB-5083 as well as with the proteasome inhibitor MG-132 for 16 hours (Fig 1). Inhibition of VCP induced a robust accumulation of ubiquity-lated protein species in a dose-dependent manner in THP-1 (Fig 1A) as well as in a long-term culture of primary AML cells (FFM05) (Fig 1B). The accumulation of ubiquitylated protein species mirrored the effect induced by the proteasome inhibitor MG-132 (Fig 1A and 1B). We further assessed the activation of the unfolded protein response (UPR) upon inhibition of VCP in THP-1, a AML cell line with wild-type fms-like tyrosine kinase 3 (FLT3), and MV4-11, a cell line carrying an FLT3-internal tandem duplication (ITD) mutation. Immunoblotting revealed an increased expression of the UPR transducers immunoglobulin heavy chain-binding protein (BiP) (also known as 78 kDa glucose-regulated protein or GRP78), inositol-requiring protein 1α (IRE1α), protein kinase R (PKR)-like endoplasmic reticulum kinase (PERK), and ER oxidoreductase 1 (Ero1)-α (Fig 1C). The levels of calnexin remained unchanged and CCAAT/enhancer-binding protein (C/EBP)-homologous protein (CHOP), an apoptotic marker, was detected neither in THP-1 nor in MV4-11 cells. Overall, these results show a potent inhibition of the ubiquitin-proteasome system and an activation of the unfolded protein response in AML cells treated with the VCP inhibitor CB-5083.

### VCP inhibition by CB-5083 alters the ubiquitin-modified proteome of AML cells

In order to characterize the effects of VCP inhibition by CB-5083 on the ubiquitin-modified proteome of AML cells and to uncover its substrates, we employed quantitative mass spectrometry-based proteomics. To this end, THP-1 and MV4-11 cells were metabolically labelled with amino acids incorporating stable isotopes in cell culture (SILAC). For the analysis of the ubiquitin-modified proteome, light labelled cells were treated with DMSO and heavy labelled cells were treated with 200 nM CB-5083 for 6 hours. Subsequently, proteins from both cell pools were isolated, mixed and digested into peptides using trypsin. Ubiquitin-modified peptides were enriched using diGly-specific antibodies and subjected to liquid chromatography and tandem mass spectrometry analysis (LC-MS/MS). Data analysis was performed using the MaxQuant software package.

The data demonstrated an increased ubiquitylation of diverse proteins including VCP itself, its adaptors (FAF1), proteasome components (PSMA3, PSMC3), autophagy-related proteins (AMBRA1, SQSTM1), transmembrane proteins (ANKRD13A, SMIM14) and DNA damage response factors (DDB1, WRNIP1) (Fig 2A and 2B, S1 Table).

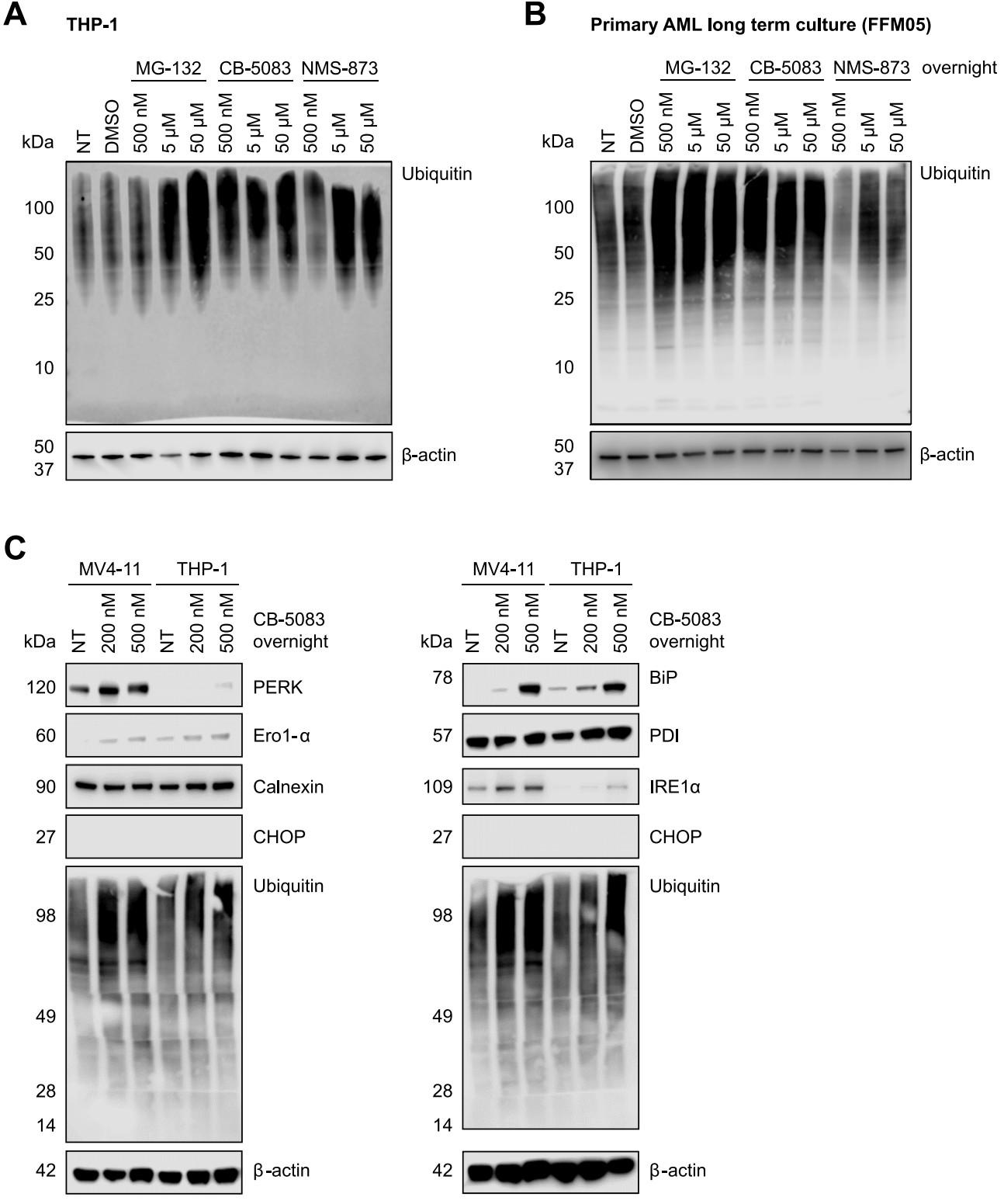

**Fig 1. Inhibition of VCP leads to accumulation of ubiquitylated proteins and activates the unfolded protein response (UPR) in human acute myeloid leukemia (AML) cells.** Immunoblot showing poly-ubiquitylated proteins in THP-1 cells (A) and in a primary human AML long-term culture (FFM05) (B) after treatment with the VCP inhibitors CB-5083 or NMS-873 or the proteasome inhibitor MG-132. (C) Immunoblot showing expression of UPR-associated proteins in THP-1 and MV4-11 cells after treatment with 200 or 500 nM CB-5083 overnight.

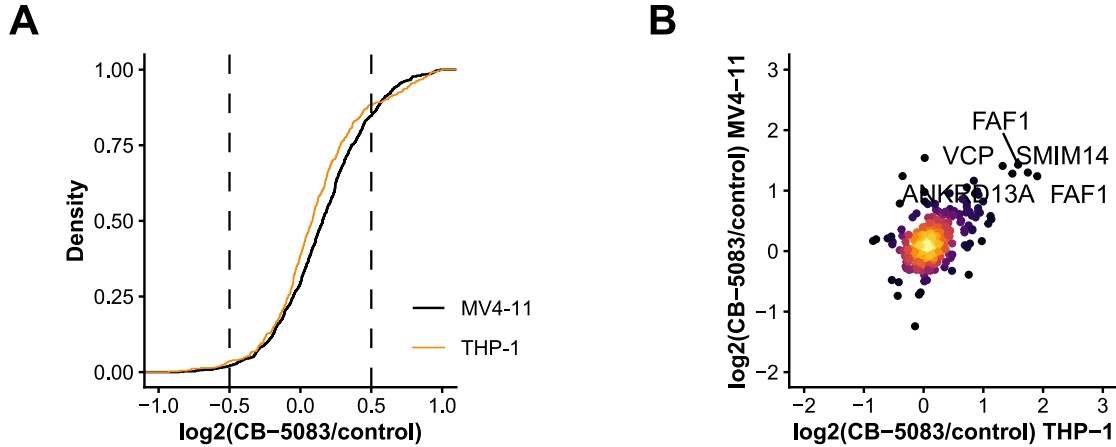

**Fig 2. Dynamics of the ubiquitin-modified proteome in AML cells after VCP inhibition.** (A) SILAC ratio distribution of all quantified ubiquitylation sites in MV4-11 and THP-1 cells. (B) Scatter plot showing SILAC ratios of quantified ubiquitylation sites in MV4-11 and THP-1 cells after treatment with the VCP inhibitor CB-5083.

## VCP inhibition with CB-5083 impairs growth and induces apoptosis in AML cells

Subsequently, we tested whether short-term inhibition of VCP can induce apoptosis of AML cells. We employed a dual staining with DNA-binding compound 7-aminoactinomycin D (7-AAD) and phosphatidylserine-binding protein annexin V with flow cytometry-based read-out. Similar to $H_2O_2$ used as a positive control, overnight treatment with 1 µM CB-5083 led to pronounced apoptosis in MV4-11 cells (Fig 3A and 3B). No apoptosis was observed in cells treated overnight with 200 nM CB-5083. In sum, higher concentrations of CB-5083 are capable of inducing an anti-leukemic effect via the apoptotic pathway.

## CB-5083 impairs AML cell growth

Next, we asked whether VCP inhibition using the specific inhibitors CB-5083 and NMS-873 affects proliferation and survival of AML cells. To this end, cells were seeded in 96-well plates with increasing concentrations of CB-5083 or NMS-873, and cell viability was assessed after 72 hours using a fluorimetric assay based on resazurin reduction. The $IC_{50}$ values of CB-5083 in the AML cell lines THP-1, MV4-11 and KG-1 ranged from 208 to 281 nM, whereas the $IC_{50}$ values of NMS-873 were consistently one order of magnitude higher, ranging from 1.25 to 1.60 µM in the tested cell lines (Fig 4A and 4D). The long-term culture of primary AML cells (FFM05) was slightly less sensitive to VCP inhibition with an $IC_{50}$ value of 528 nM for CB-5083 and 3.37 µM for NMS-873 (Fig 4A and 4D).

To assess potential effects of VCP inhibition on normal cells, we also determined the $IC_{50}$ values for the VCP inhibitors in hematopoietic progenitors (CD34-positive cells isolated from healthy donors) and in the human bone marrow-derived stromal cell line HS-5. In CD34-positive cells isolated from healthy donors the $IC_{50}$ for CB-5083 ranged from 353 to 374 nM (Fig 4B and 4C). The $IC_{50}$ value for CB-5083 in HS-5 cells was 356 nM (Fig 4C and 4D).

Taken together, CB-5083 exerts a dose-dependent effect on proliferation and survival in different AML cell lines. However, we also observed similar $IC_{50}$ values for hematopoietic progenitors and bone marrow stromal cells.

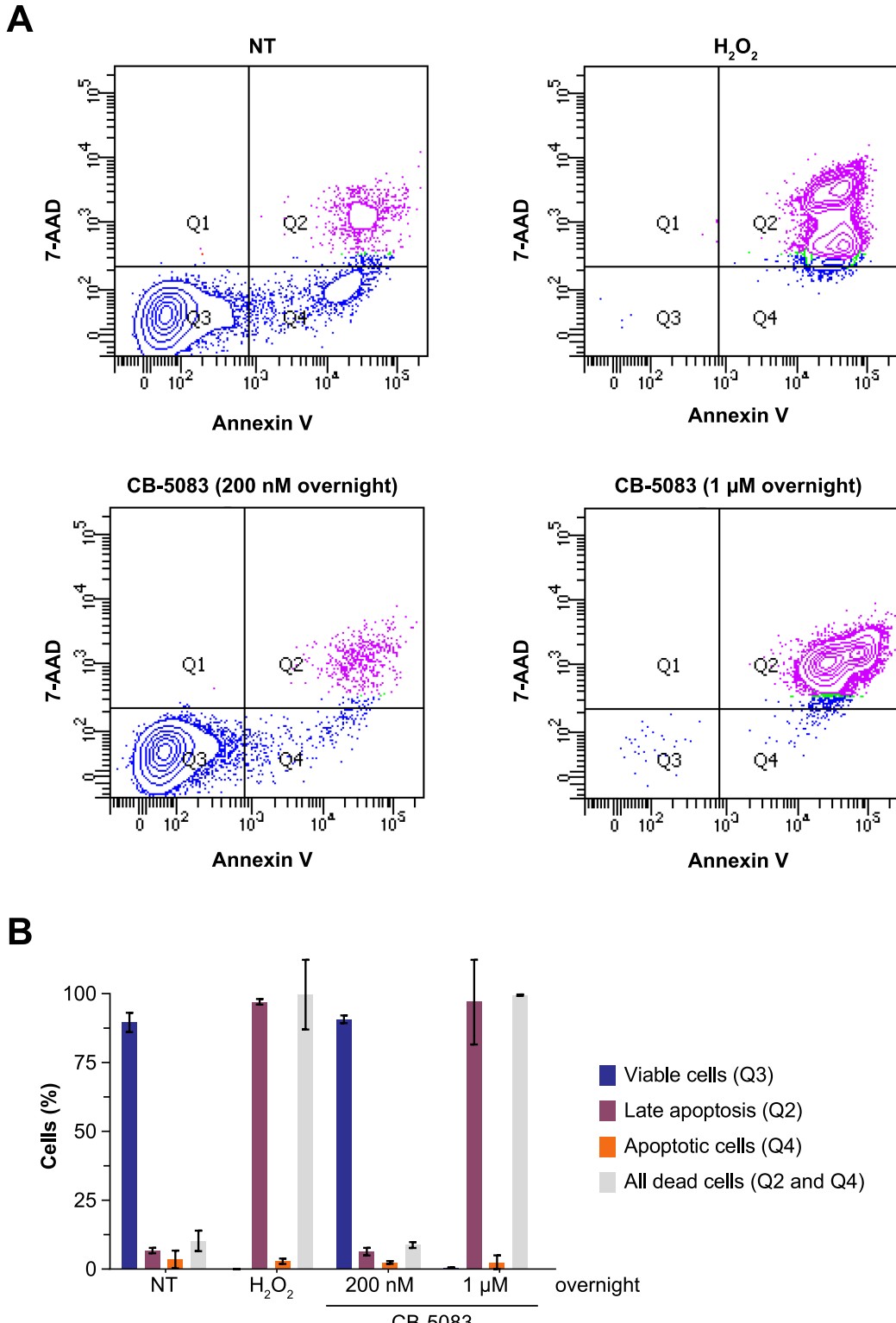

**Fig 3. Inhibition of VCP induces apoptosis in AML cells.** (A) MV4-11 cells were treated with 200 nM or 1 μM CB-5083 overnight. Cells that were left untreated or treated with 500 mM $H_2O_2$ served as control. Induction of apoptosis was assessed by dual staining with Annexin V and 7-AAD. Relative fluorescence of Annexin V and 7-AAD is shown for all conditions. (B) Quantification of the data.

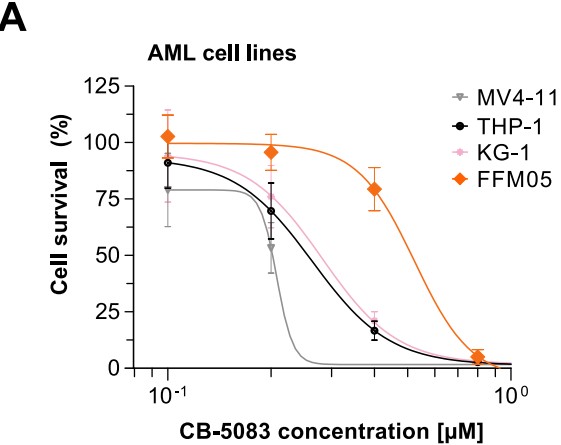

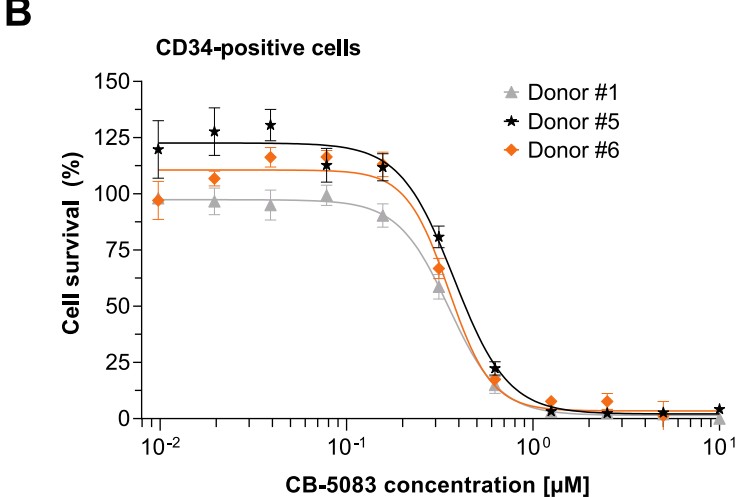

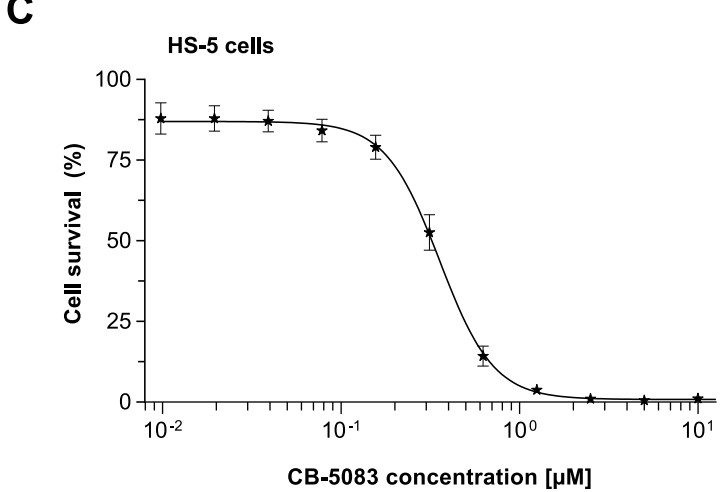

**D**

| Cell type | IC$_{50}$ (nM) CB-5083 | IC$_{50}$ (µM) NMS-873 |
|---|---|---|
| MV4-11 | 208 | 1.60 |
| THP-1 | 265 | 1.25 |
| KG-1 | 281 | 1.25 |
| FFM05 | 528 | 3.37 |
| CD34-positive (#1) | 353 | ND |
| CD34-positive (#5) | 374 | ND |
| CD34-positive (#6) | 354 | ND |
| HS-5 | 356 | 3.39 |

**Fig 4. VCP inhibition reduces AML cell growth and survival.** (A) AML cell lines THP-1, MV4-11, and KG-1 as well as the primary AML long term culture FFM05 were treated with different concentrations of CB-5083 and NMS-873. Cell count was measured after 72 hours and IC$_{50}$ values for all cell lines were estimated. Additionally, IC$_{50}$ values for CB-5083 were determined for CD34-positive cells from donors (B) and for the human bone marrow stromal cell line HS-5 (C). (D) Summary of all determined IC$_{50}$ values.

## CB-5083 and cytarabine synergistically decrease AML cell survival

The pyrimidine nucleoside analogue cytosine β-D-arabinofuranoside (cytarabine) is one of the most widely used drugs for the therapy of patients with AML. Cytarabine is also frequently combined with other cytostatic drugs or with specific inhibitors such as venetoclax to yield synergistic effects [20, 21]. We set out to investigate whether there is a synergistic effect of the VCP inhibitor CB-5083 and cytarabine. To this end, AML cell lines THP-1 and MV4-11 AML were seeded into 96-well plates with different concentrations of CB-5083 and cytarabine. Cell viability was assessed after 72 hours as described above and the data from the cell viability assays were used as input for the SynergyFinder web tool in order to generate 2D and 3D synergy landscape maps [22]. A positive value of the δ score indicates synergy (Fig 5); the ZIP model was used as a reference model (the expected response corresponds to an additive effect as if the two drugs do not affect the potency of each other). The experiments showed a δ score of 7.023 for THP-1 and 8.189 for MV4-11 cells, indicating that the two drugs (CB-5083 and cytarabine) synergistically affect proliferation and survival of the tested cell lines.

In addition to cytarabine, we evaluated the BH3-mimetic venetoclax as a potential combination partner for CB-5083. The BH3-mimetic venetoclax blocks the anti-apoptotic B-cell lymphoma-2 (Bcl-2) protein and thereby induces apoptosis in cancer cells. Recently, clinical trials have demonstrated the efficacy of venetoclax for the treatment of patients with AML [21, 23]. We showed that VCP inhibition can lead to induction of apoptosis in AML cells (Fig 3). Therefore, we also set out to investigate if the VCP inhibitor CB-5083 and the Bcl-2 inhibitor Venetoclax synergistically reduce viability in AML cells. As in case of CB-5083/cytarabine, we could observe a synergistic effect of CB-5083/venetoclax on AML cell proliferation/survival with a more pronounced effect in THP-1 cells (Fig 5C and 5D).

## Prolonged treatment of AML cells with CB-5083 leads to development of resistance

We further tested whether prolonged treatment of AML cells can lead to resistance against inhibition of VCP by CB-5083. To this end, MV4-11 cells were adapted stepwise to a growth in presence of up to 1 μM CB-5083 over the course of three months. To assess the activation of the UPR in cells consistently exposed to CB-5083 we monitored the expression of UPR markers in MV4-11 cells grown in absence and in presence of 200 or 500 nM CB-5083 (Fig 6A). The expression of BiP, protein disulfide-isomerase (PDI), IRE1α and PERK was increased in cells grown in presence of CB-5083 and correlated with the concentration of CB-5083 in the media. We also detected an increased amount of polyubiquitylated protein species in cells grown in the presence of CB-5083. The levels of calnexin remained unchanged whereas those of Ero1-α were slightly lower. Of note, CCAAT/enhancer-binding protein (C/EBP)-homologous protein (CHOP), an apoptotic marker, was detected in neither parental nor in CB-5083-resistant cells. An analysis of apoptosis using a 7-AAD/annexin V staining revealed no induction of apoptosis in MV4-11 cells grown in the presence of 1 μM CB-5083 (Fig 6B and 6C).

After establishment of the resistant cell lines, we withdrew CB-5083 from the media for two days and performed a 72-hour cell survival assay in order to determine the $IC_{50}$ of CB-5083 in these cells. The $IC_{50}$ values in CB-5083-resistant cells were consistently higher than in the parental cell line, ranging from 407 nM for cells grown in presence of 500 nM CB-5083 to ~5 μM in cells grown in presence of 1 μM CB-5083 (Fig 6D and 6E).

To detect whether the CB-5083-resistant AML cells might harbor mutations in *VCP*, we isolated genomic DNA, amplified the relevant DNA regions by PCR and analyzed the sequence by Sanger DNA sequencing. This analysis could identify a homozygous single

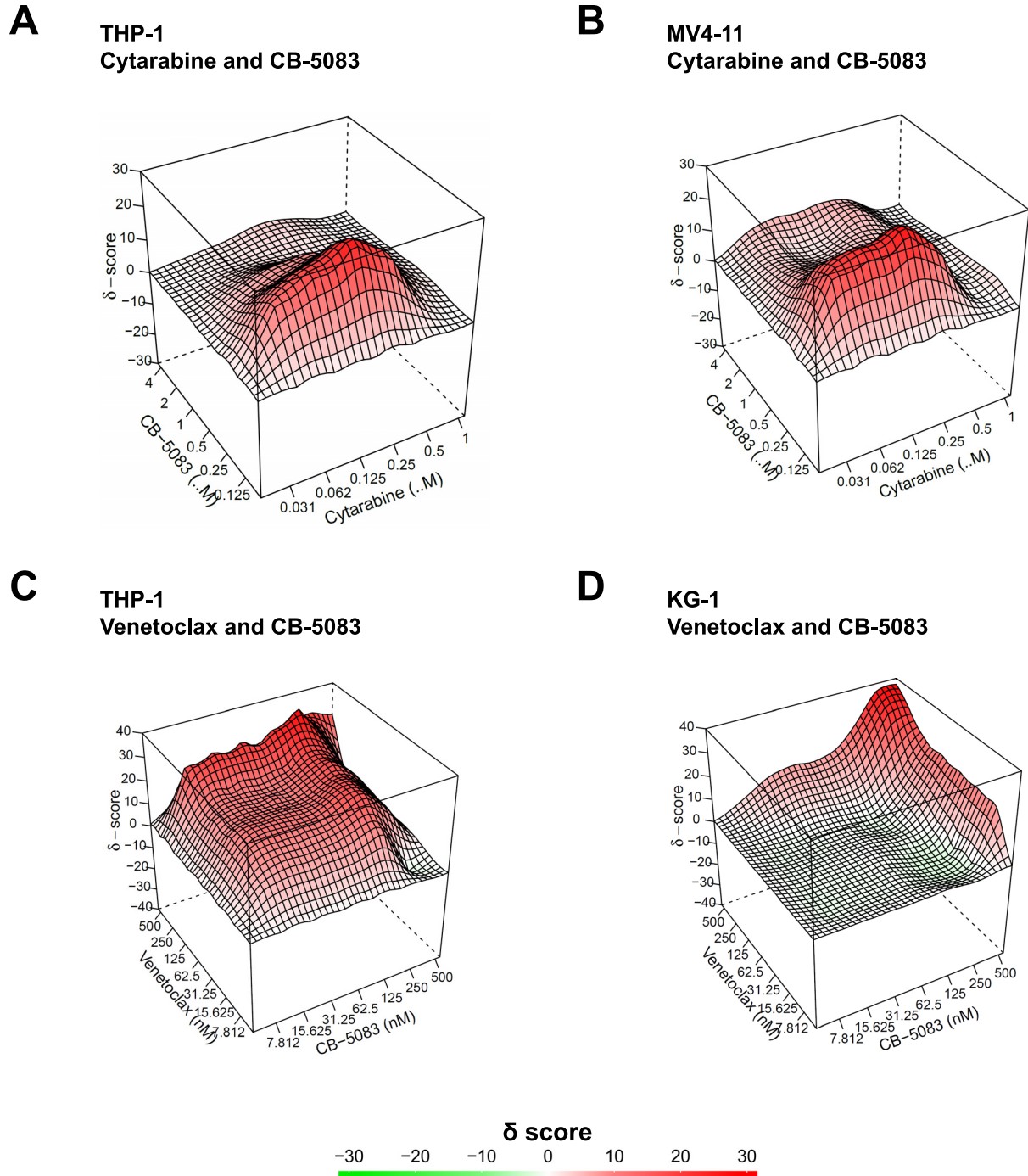

**Fig 5. CB-5083 and cytarabine or venetoclax synergistically decrease AML cell survival.** (A-D) THP-1 and MV4-11 cell survival was assessed in a 96-well format with increasing concentrations of CB-5083 and cytarabine or venetoclax. Following a 72-hour incubation, CellTiter-Blue Reagent was added and fluorescence/cell viability was assessed. Cell viability data was used as input for the SynergyFinder software to estimate synergy δ scores. Synergy landscape maps display δ scores for the indicated cells lines and drug combinations.

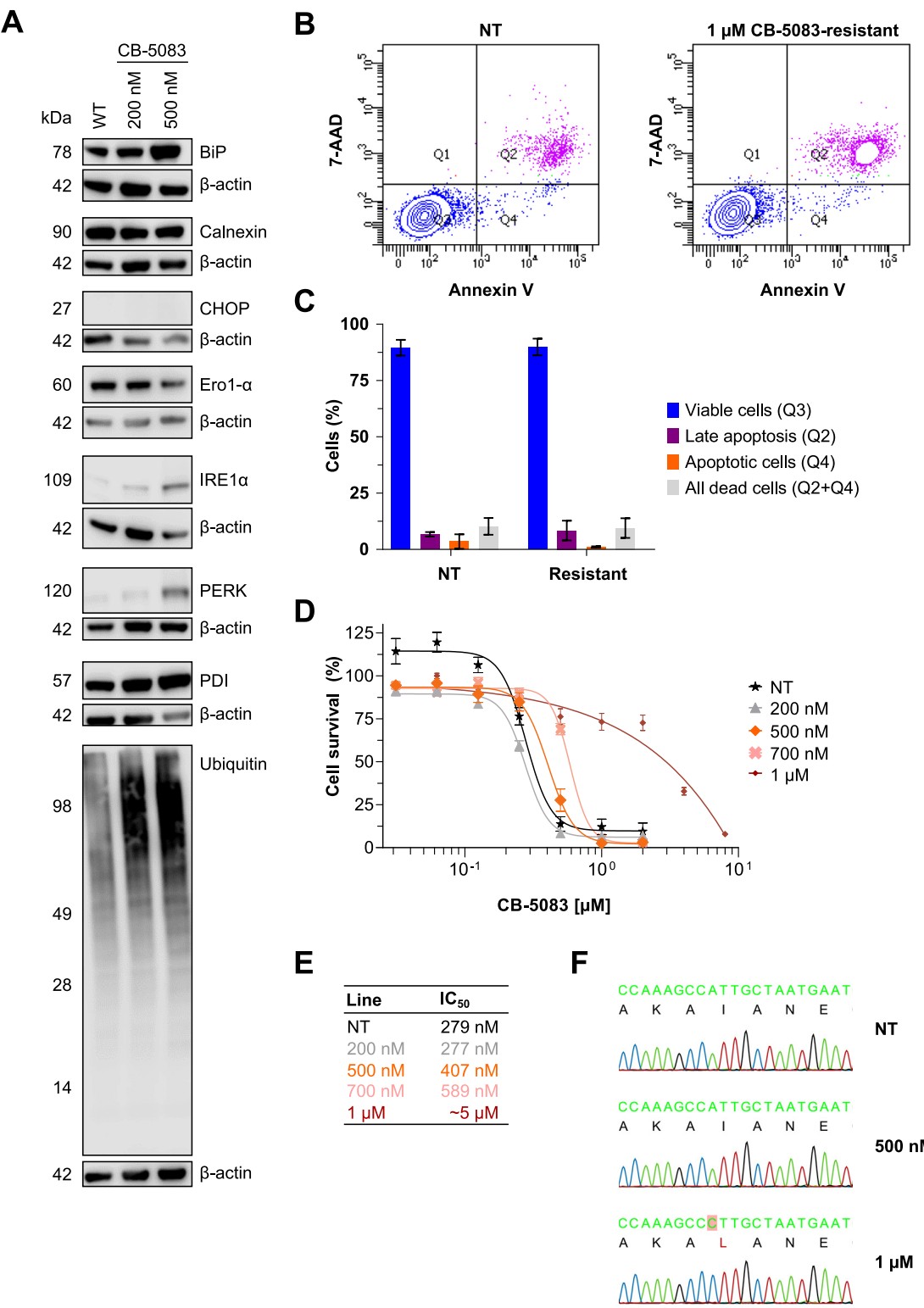

**Fig 6. CB-5083-resistant MV4-11 cells exhibit features of permanent endoplasmic reticulum (ER) stress but no apoptosis.**
(A) Immunoblot showing induction of ER stress in inhibitor-resistant cells after a few months of continuous treatment with 200 and 500 nM CB-5083. (B and C) Flow cytometry analysis using Annexin V and 7-AAD staining showed decreased apoptosis in CB-5083-resistant cell lines. (D and E) IC$_{50}$ values for CB-5083 in the indicated cell lines upon withdrawal and re-treatment with CB-5083. (F) Analysis of the coding sequence of VCP identified a variant (c.1591C>A p.I531L) in the cell line treated with 1 μM CB-5083.

nucleotide variant in exon 13 of VCP (c.1591C>A p.I531L) in cells grown in presence of 1 μM CB-5083 (Fig 6F). The identified variant is located in the second ATPase domain (D2) of VCP.

## Discussion

Treatment of patients with AML remains a challenge and novel therapies are urgently needed. In a targeted shRNA-based genetic screen for UPS-related genes required for AML cell survival, we identified VCP as a candidate gene.

In this study we employed the specific VCP inhibitor CB-5083 to demonstrate that inhibition of VCP in AML cells leads to accumulation of ubiquitylated proteins, activation of the UPR and reduces survival of multiple AML cells lines. Previous studies have indicated that VCP inhibition decreased the survival of multiple cancer cell lines: CB-5083 has been reported to diminish cell survival of human multiple myeloma [11], human acute lymphoblastic leukemia [12] and human mantle cell lymphoma [14]. A recent study also investigated the effect of VCP inhibition on AML cells [15].

The mechanisms how inhibition of VCP affects AML cell survival remain incompletely understood. To gain further insights into the mechanism of action we employed quantitative mass spectrometry-based proteomics to identify substrates of VCP in AML cells. Our results reveal that VCP targets multiple components of the ubiquitin-machinery as well as proteins involved in autophagy and DNA damage repair/response. These findings are in line with a recently published study that links VCP to the cellular DNA repair in AML cells [15].

We further demonstrate that combinations of the VCP inhibitor CB-5083 with the antimetabolite cytarabine or the Bcl-2 inhibitor venetoclax act synergistically in killing different AML Interestingly, also another recently published study highlights a synergistic effect of VCP and Bcl-2 inhibition [15].

We also investigated if prolonged treatment of AML cells with CB-5083 can induce development of resistance: Aim of the unfolded protein response is to restore ER protein homeostasis however prolonged unfolded protein response can lead to induction of apoptosis. CHOP is considered a central regulator of ER stress induced apoptosis and its expression can be induced by the PERK–eIF2α–ATF4 as well as IRE1α–TRAF2 signaling axis. Interestingly, we observed reduced levels of PERK and IRE1α in cells that have been exposed to the VCP inhibitor CB-5083 for extended time periods (Figs 1C and 6A). The reduced expression levels of these proteins might be a result of the adaptation to persistent ER stress and potentially prevent induction of apoptosis. At concentrations above 1 μM we observed acquisition of a resistance mutation in the ATPase domain of VCP (p.I531L). Resistance-conferring mutations in the D1 and D2 region of VCP have been described before [9, 23, 24] but to our knowledge the resistance mutation reported in this study has not been described before (Table 1).

**Table 1. Potential CB-5082 resistance mutations characterized in this and previous studies.**

| Study | Cancer type (cell line) | Variants (domain) |
|---|---|---|
| This study | Acute myeloid leukemia (MV4-11) | p.I531L (D2) |
| Bastola et al. (2017) | Ovarian cancer (OVSAHO) | p.E470K (D1-D2 linker)<br>p.E470D (D1-D2 linker)<br>p.Q603* (D2)<br>p.N616Mfs*63 (D2) |
| Anderson et al. (2015) | Colorectal cancer (HCT116) | p.P472L (D1-D2 linker)<br>p.Q473P (D1-D2 linker)<br>p.V474A (D1-D2 linker)<br>p.N660K (D2)<br>p.T688A (D2) |
| Wei et al. (2018) | Colorectal cancer (HCT116) | p.P472L (D1-D2 linker) |

In summary, we demonstrate that inhibition of VCP in AML cells induces a potent unfolded protein response and apoptosis and decreases cell proliferation. We observe synergistic effects in cells treated with CB-5083 and cytarabine as well as with CB-5083 and the Bcl-2 inhibitor venetoclax. These results suggest that inhibition of VCP alone or in combinations with other drugs might serve as a therapeutic strategy in patients suffering from AML. Clinical development of CB-5083 was terminated because an off-target effect resulting in vision problems has been observed (NCT02243917 and NCT02223598). However, an improved molecule developed by the same company is currently in phase I trials for the treatment of AML, MDS and solid cancers (NCT04402541 and NCT04372641). We also observed that VCP inhibition affects proliferation of CD34-positive hematopoietic progenitors and that prolonged treatment with the VCP inhibitor CB-5083 induces development of resistance, possibly limiting the clinical utility of VCP inhibitors.

## Material and methods

### Inhibitors

CB-5083, NMS-873 and MG-132 were purchased from Selleck Chemicals, cytosine β-D-arabinofuranoside (cytarabine) was purchased from Sigma-Aldrich. All inhibitors were dissolved in dimethyl sulfoxide (Sigma-Aldrich).

### Antibodies

The following primary antibodies were used (dilution 1:1,000): mouse monoclonal ubiquitin antibody (P4D1) (cat. no. sc-8017, Santa Cruz); rabbit monoclonal β-actin antibody (D6A8) (cat. no. 8457, Cell Signalling); ER stress sampler kit (cat. no. 9956, Cell Signalling) containing BiP (C50B12) (cat. no. 3177), calnexin (C5C9) (cat. no. 2679), Ero1-α (cat. no. 3264), IRE1α (cat. no. 3294), CHOP (L63F7) (cat. no. 2895), PERK (D11A8) (cat. no. 5683), and PDI antibody (C81H6) (cat. no. 3501). The following secondary antibodies were used (dilution 1:10,000): peroxidase AffiniPure F(ab')$_2$ fragment goat anti-rabbit IgG (H+L) and peroxidase AffiniPure F(ab')$_2$ fragment goat anti-mouse IgG (H+L) (cat. no. 111-036-003 and 115-036-003, Jackson ImmunoResearch Laboratories).

For flow cytometry, dye coupled antibodies CD11b-APC (cat. no. 130-098-088, BD), CD34-PE (8G12) (cat. no. 345802, BD), and CD45-V450 (cat. no. 642275, BD) were used for direct detection (dilution 1:100).

### Cell culture

Acute monocytic leukemia cell lines THP-1 and MV4-11 were obtained from DSMZ and bone marrow stromal cell line HS-5 from ATCC. Cell culture reagents were obtained from Thermo Fisher Scientific unless otherwise stated.

Acute myeloid leukemia (THP-1 and MV4-11) were maintained in RPMI 1640 medium supplemented with 10% (v/v) heat-inactivated FBS, 2 mM L-glutamine, penicillin and streptomycin in a humidified incubator at 37˚C and 5% $CO_2$. Establishment of FFM05 cells has been previously described [25]. FFM05 cells were maintained in X-Vivo medium (Lonza) supplemented with 10% (v/v) FBS HyClone, 2 mM L-glutamine, and human recombinant growth factory: thrombopoietin (TPO) (25 ng/ml), stem cell factor (SCF) (50 ng/ml), FMS-related tyrosine kinase 3 ligand (FLT3-I) ligand (50 ng/ml), and interleukin-3 (IL-3) (20 ng/ml) (Miltenyi Biotech).

## Donor samples and CD34[+] mononuclear cell enrichment

CD34+ mononuclear cells were isolated from peripheral blood samples after G-CSF stimulation. Use of the samples for research purposes was approved by the Ethics Committee of the University of Frankfurt (statement 329–10) and donors gave written consent for use of the samples. For purification, cell samples were diluted with PBS and carefully transferred on top of Ficoll-Paque (cat. no. 17-5442-02, GE Healthcare). Following a gradient centrifugation at 400×*g* for 30 min. at room temperature, the upper plasma layer was removed, and the interphase was collected for CD34[+] cell enrichment using the CD34 MicroBead Kit (cat. no. 130-046-702, Miltenyi Biotech) according to the manufacturer's instructions. After magnetic separation, cells were centrifuged and resuspended in complete X-Vivo medium.

For assessment of purity, cells were stained with fluorescent dye coupled antibodies (CD11b, CD34 and CD45 antibody) and Fixable Viability Dye eFluor 780 (cat. no. 65-0865-14, eBioscience). Flow cytometry analysis was performed on a LSRFortessa flow cytometer (BD Biosciences) and data was analyzed using the FACSDiva software (BD Biosciences).

After assessment of purity, the cells were seeded in 96-well plates for survival assays (see below).

## Apoptosis assays

Apoptosis assays were performed using the FITC Annexin V Apoptosis Detection Kit with 7-AAD (cat. no. 640922, BioLegend). To this end, 1 million cells were washed with PBS/2% FCS, centrifuged at 1,200 rpm for 5 min. at room temperature and resuspended in 400 µl Annexin V Binding Buffer (AVBP). Subsequently, samples were divided into 4 aliquots. One aliquot served as an unstained control, and the other aliquots were combined with either FITC Annexin V, 7-AAD, or both. Following incubation for 15 min. at room temperature, 400 µl AVBP was added and the cells were analyzed on the flow cytometer. The following emission filters were used: 7-AAD G-610/20, Annexin V B-530/30. Compensation was enabled for the channel G-610/20 minus % B-530/30 (spectral overlap 0.30%).

## Cell survival assays, IC$_{50}$ determination and synergy calculation

Cell viability was assessed using the CellTiter-Blue Cell Viability Assay (Promega) according to manufacturer's instructions. Briefly, cells were seeded in triplicate in 96-well plates and inhibitors were added at the indicated concentrations and incubated for 72 hours. Fluorescence was measured 4 h after the addition of the CellTiter -Blue Reagent. Experiments were repeated three times unless otherwise indicated.

Baseline fluorescence (medium only) was subtracted from all values, and concentration data was log-transformed in order to determine IC$_{50}$ values using GraphPad Prism. The following function was used: *log(agonist) vs. response—Find ECanything*; effective concentration (ECF) parameter F = 50.

For assessment of synergy, cells were treated with CB-5083 and cytarabine or venetoclax combinations within a 96-well plate. R-based SyngeryFinder software was used to evaluate the synergy between CB-5083 and cytarabine or venetoclax as described previously [22].

## Protein isolation and immunoblot

Cells were washed twice in PBS, lysed in modified RIPA buffer (50 mM Tris-HCl pH 7.5, 1 mM EDTA, 0.1% Na-deoxycholate, 150 mM NaCl, 1% NP-40, protease inhibitor cocktail, 5 mM β-glycerophosphate, 5 mM NaF, 1 mM Na-orthovanadate, 10 mM N-ethylmaleimide),

                                                   

and centrifuged at 15,000×*g* for 15 min. at 4˚C. Protein concentrations were estimated using Bradford Reagent (AppliChem).

Protein extracts were combined with 4× Laemmli Sample Buffer (Bio-Rad), boiled for 10 min at 70˚C, resolved on MiniPROTEAN TGX pre-cast 4–15% SDS-polyacrylamide gels (10- or 15-well), and transferred onto Hyperfilm enhanced chemiluminescence (ECL) nitrocellulose membranes (GE Healthcare). Following the transfer, membranes were blocked with 5% milk in TBST, and probed with primary and secondary antibodies. For visualization Super-Signal West Femto Maximum Sensitivity ECL Substrate (Thermo Fisher Scientific) and an Odyssey imaging system (LI-COR Biosciences) were used.

## Mass spectrometry-based proteomics

SILAC (stable isotope labelling with amino acids in cell culture) was performed as described previously [26]. Briefly, cells were washed and resuspended in light and heavy SILAC media— arginine/lysine-free RPMI supplemented with dialysed FCS (lot no. 14J422, cat. no. F0392, Sigma-Aldrich). The following light/heavy isotopes of amino acids were used: arginine-0 (light) (cat. no. A6969, Sigma-Aldrich); arginine-10 (heavy) (cat. no. CNLM-539-H-1, Cambridge Isotope Laboratories); lysine-0 (light) (cat. no. L8662, Sigma-Aldrich); and lysine-8 (heavy) (cat. no. CNLM-291-H-1, Cambridge Isotope Laboratories). Cells were then cultured in 6-well plates for approximately 4 weeks. For assessment of proteome dynamics after VCP inhibition, cells were treated with 200 nM CB-5083 or DMSO for 6 (ubiquitylome analysis) or 24 h (proteome analysis), washed, centrifuged, and lysed in a modified RIPA buffer.

Ubiquitylome analysis was performed as described previously [27]: proteins were precipitated in fourfold excess of ice-cold acetone and subsequently re-dissolved in denaturation buffer (6 M urea, 2 M thiourea in 10 mM HEPES pH 8.0). Cysteines were reduced with 1 mM dithiothreitol and alkylated with 5.5 mM chloroacetamide [28]. Proteins were digested with endoproteinase Lys-C (Wako Chemicals) and sequencing grade modified trypsin (Sigma). Protease digestion was stopped by addition of trifluoroacetic acid to 0.5%, and precipitates were removed by centrifugation. Peptides were purified using reversed-phase Sep-Pak C18 cartridges (Waters) and eluted in 50% acetonitrile. For ubiquitin remnant peptide enrichment, 20 mg of peptides was re-dissolved in immunoprecipitation buffer (10 mM sodium phosphate, 50 mM sodium chloride in 50 mM MOPS pH 7.2). Precipitates were removed by centrifugation. Modified peptides were enriched using 40 µl of di-glycine-lysine antibody resin (Cell Signaling Technology). Peptides were incubated with the antibodies for 4 h at 4˚C on a rotation wheel. The beads were washed three times in ice-cold immunoprecipitation buffer followed by three washes in water. The enriched peptides were eluted with 0.15% trifluoroacetic acid in water, fractionated in six fractions using micro-column-based strong-cation exchange chromatography (SCX) [29], and desalted on reversed-phase C18 StageTips [30].

Peptide fractions were analyzed on a quadrupole Orbitrap mass spectrometer (Q Exactive Plus, Thermo Fisher Scientific) equipped with a UHPLC system (EASY-nLC 1000, Thermo Fisher Scientific) as described previously [31]. Peptide samples were loaded onto C18 reversed-phase columns (15 cm length, 75 µm inner diameter, 1.9 µm bead size) and eluted with a linear gradient from 8 to 40% acetonitrile containing 0.1% formic acid in 2 h. The mass spectrometer was operated in data-dependent mode, automatically switching between MS and MS2 acquisition. Survey full scan MS spectra (m/z 300–1,700) were acquired in the Orbitrap. The 10 most intense ions were sequentially isolated and fragmented by higher energy C-trap dissociation (HCD) [32]. An ion selection threshold of 5,000 was used. Peptides with unassigned charge states, as well as with charge states < +2, were excluded from fragmentation. Fragment spectra were acquired in the Orbitrap mass analyzer.

Raw data files were analyzed using MaxQuant (version 1.6.17.0) [33]. Parent ion and MS2 spectra were searched against a database containing 88,473 human protein sequences, human protein sequences obtained from the UniProtKB using Andromeda search engine. Spectra were searched with a mass tolerance of 6 ppm in MS mode, 20 ppm in HCD MS2 mode, strict trypsin specificity and allowing up to three miscleavages. Cysteine carbamidomethylation was searched as a fixed modification, whereas protein N-terminal acetylation, methionine oxidation, N-ethylmaleimide modification of cysteines (mass difference to cysteine carbamidomethylation), and di-glycine-lysine were searched as variable modifications. Site localization probabilities were determined by MaxQuant using the PTM scoring algorithm as described previously. The dataset was filtered based on posterior error probability (PEP) to arrive at a false discovery rate of below 1% estimated using a target-decoy approach [34]. Di-glycine lysine-modified peptides with a minimum score of 40 and delta score of 6 are reported and used for the analyses.

Secondary statistical analysis was performed using the R software environment (version 4.1.1).

## Generation of resistant cell lines

For generation of CB-5083-resistant cells, MV4-11 cells were cultured with increasing concentration of CB-5083 (200 nM to 1.5 μM). For the analysis of the VCP sequence, genomic DNA was extracted from cell pellets using the Blood & Cell Culture DNA Midi Kit (cat. no. 13343, Qiagen). Coding sequences of VCP were PCR-amplified using the primers listed below. The PCR products were cleaned up using the NucleoSpin Gel and PCR Clean-up Kit (cat. no. 740609.250, Macherey-Nagel) and analysed by Sanger sequencing using M13 primers (Microsynth Seqlab). DNA sequence alignment was performed with CLC Sequence Viewer (Qiagen).

| Name | Forward primer sequence | Reverse primer sequence |
|---|---|---|
| VCP-Ex-11_12 | TGTAAAACGACGGCCAGTGGGTCTTTGAGGCAGCATA | CAGGAAACAGCTATGACTGACTCACCCTGGACCAAGT |
| VCP-Ex-13 | TGTAAAACGACGGCCAGTAATGGAGGGGATGCTTCTG | CAGGAAACAGCTATGACGCCCTCAGGCAAATCAATAC |
| VCP-Ex-14 | TGTAAAACGACGGCCAGCATGCTGGTTTCGGATTTCT | CAGGAAACAGCTATGACGCCTGAGGACTCATGCAAGT |
| VCP-Ex-15 | TGTAAAACGACGGCCAGGGGTTGGTCTAAAGGGAAGG | CAGGAAACAGCTATGACTCTCCATGATTGGCACATCT |
| VCP-Ex-16 | TGTAAAACGACGGCCAGTTTCCAGAGTGCATTGACAAGT | CAGGAAACAGCTATGACTTTGGTGTAGGTCCCCAAAG |

## Supporting information

**S1 Table. Quantified ubiquitylation sites.**
(XLSX)

**S1 Raw images. Raw western blot images.**
(PDF)

## Acknowledgments

The authors thank all lab members for constructive discussions.

## Author Contributions

**Conceptualization:** Sebastian A. Wagner.

**Data curation:** Jan B. Heidelberger.

**Formal analysis:** Paweł P. Szczęśniak.

**Funding acquisition:** Sebastian A. Wagner.

**Methodology:** Jan B. Heidelberger.

**Resources:** Hubert Serve.

**Writing – original draft:** Paweł P. Szczęśniak, Petra Beli, Sebastian A. Wagner.

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
