## [Decision Letter · Decision Letter 0]

28 Dec 2021

PONE-D-21-30344VCP inhibition induces an unfolded protein response and apoptosis in human acute myeloid leukemia cellsPLOS ONE

Dear Dr. Wagner,

Thank you for submitting your manuscript to PLOS ONE. After careful consideration, we feel that it has merit but does not fully meet PLOS ONE’s publication criteria as it currently stands. Therefore, we invite you to submit a revised version of the manuscript that addresses the points raised during the review process.

We look forward to receiving your revised manuscript.

Kind regards,

Anilkumar Gopalakrishnapillai

Academic Editor

PLOS ONE

Journal Requirements:

The study was supported by the ProLOEWE Ubiquitin Networks (Ub-Net) research cluster of the State of Hessen and by a project grant from the Else Kröner-Fresenius Foundation awarded to SW (2015_A124). 

The study was supported by the ProLOEWE Ubiquitin Networks (Ub-Net) research cluster of the State of Hessen and by a project grant from the Else Kröner-Fresenius Foundation awarded to SW (2015_A124).

The study was supported by the ProLOEWE Ubiquitin Networks (Ub-Net) research cluster of the State of Hessen and by a project grant from the Else Kröner-Fresenius Foundation awarded to SW (2015_A124).

Reviewers' comments:

Reviewer's Responses to Questions

**Comments to the Author**

1. Is the manuscript technically sound, and do the data support the conclusions?

Reviewer #1: Yes

Reviewer #2: Yes

2. Has the statistical analysis been performed appropriately and rigorously? 

Reviewer #1: Yes

Reviewer #2: Yes

3. Have the authors made all data underlying the findings in their manuscript fully available?

Reviewer #1: Yes

Reviewer #2: Yes

4. Is the manuscript presented in an intelligible fashion and written in standard English?

Reviewer #1: Yes

Reviewer #2: Yes

5. Review Comments to the Author

Reviewer #1: The authors performed a series of in vitro experiments to demonstrate the effects of VCP

( valosin-containing protein ) inhibitor CB-5083 on the human leukemia cell lines THP-1 and

MV4-11. They have successfully showed that CB-5083 impaired the cell growth and viability

of these leukemia cells and promoted cellular apoptosis via accumulation of ubiquitylated

proteins and activation of unfolded protein response (UPR). They have also observed that the

combination of VCP inhibitor and cytarabine or the Bcl-2 inhibitor venetoclax may exert

synergistic effects on these leukemia cell lines. Despite the leukemia cell may develop resistance

to CB-5083 by VCP gene mutation after a longer exposure , the VCP inhibitor, especially in

combination with other agents, still provide a promising therapeutic strategy for treatment of

AML patients.

This study has been elaborately designed and carefully conducted. All techniques explored in

this study are described in detail. The data presented in figures and tables are clear, sound and

persuasive. My comments and suggestions are as the follows:

1, As shown in Fig 5, CB5083 combined with cytarabine or venetoclax synergistically impaired

the growth and reduced the viability of AML cells, but no data support if these combinations can

induce apoptosis of AML cells in synergy. Did these combinations do so?

2, If the combination of CB5083 with cytarabine or venetoclax did synergistically induce

apoptosis of the AML cells, could it be an effective way to overcome the resistance to CB5083?

3, Compare the Western blot results in Fig1C to that in Fig 6. In Fig1C the amount of Ero-1α

increased as the MV4-11 treated with CB5083 at 200nM increased to 500nM, while in Fig 6 the

high level of Ero-1α gradually decreased in the CB5083-resistant MV4-11 treated with CB5083

from 200nM to 500nM. Could the authors explain a little bit on this issue?

4, As shown in Fig 6, the CB5083-resistant MV4-11 cells led by VCP gene mutation exhibited

permanent ER stress, but no apoptosis occurred. Please briefly discuss the possible mechanism of

this phenomena in DISCUSSION, especially on the relationship among UPR, ER stress and

apoptosis.

5, The origin of human BM stromal cell line HS-5 should be mentioned in MTERIALS AND

METHODS

I like to recommend accept this manuscript for publication after the authors revise or explain on

these minor points mentioned above.

Reviewer #2: This is an interesting study and the authors have collected a unique dataset using cutting

edge methodology. The paper is generally well written and structured. An excellent report on very thorough research. The literature review was thorough, the methodology was painstakingly thorough.

6. PLOS authors have the option to publish the peer review history of their article (what does this mean?). If published, this will include your full peer review and any attached files.

Reviewer #1: **Yes: **Zi-xing Chen

Reviewer #2: No

---

## [Author Response · Author response to Decision Letter 0]

2 Feb 2022

Response to Reviewers

Reviewer #1: The authors performed a series of in vitro experiments to demonstrate the effects of VCP (valosin-containing protein ) inhibitor CB-5083 on the human leukemia cell lines THP-1 and MV4-11. They have successfully showed that CB-5083 impaired the cell growth and viability of these leukemia cells and promoted cellular apoptosis via accumulation of ubiquitylated proteins and activation of unfolded protein response (UPR). They have also observed that the combination of VCP inhibitor and cytarabine or the Bcl-2 inhibitor venetoclax may exert synergistic effects on these leukemia cell lines. Despite the leukemia cell may develop resistance to CB-5083 by VCP gene mutation after a longer exposure, the VCP inhibitor, especially in combination with other agents, still provide a promising therapeutic strategy for treatment of AML patients.

This study has been elaborately designed and carefully conducted. All techniques explored in this study are described in detail. The data presented in figures and tables are clear, sound and persuasive. My comments and suggestions are as the follows:

We thank the reviewer for the positive evaluation of our work.

1, As shown in Fig 5, CB5083 combined with cytarabine or venetoclax synergistically impaired the growth and reduced the viability of AML cells, but no data support if these combinations can induce apoptosis of AML cells in synergy. Did these combinations do so?

Venetoclax leads to induction of apoptosis in AML cell lines at the concentrations used in our assay. A synergistic effect on apoptosis has not been investigated in our study. We agree with the reviewer that this is an interesting aspect that warrants further investigations.

2, If the combination of CB5083 with cytarabine or venetoclax did synergistically induce apoptosis of the AML cells, could it be an effective way to overcome the resistance to CB5083?

Our data provide hints that resistance to CB-5083 can be mediated by adaption of the cellular signaling via downregulation of apoptosis-inducing factors. In addition, we could also detect development of break through mutations that seem to render VCP resistant to CB-5083. Development of CB-5083 resistance mutations also seems to be the primary mechanism of development of resistance in vivo. We believe that it is unlikely that combined treatment with venetoclax can overcome resistance arising as consequence of VCP mutations. Nevertheless, venetoclax is a highly active compound in AML and there might be a possibility that combined treatment of cells with CB-5083 and venetoclax can overcome the gradual adaption of cells to increased ER stress that we observed before development of the resistance mutations.

3, Compare the Western blot results in Fig1C to that in Fig 6. In Fig1C the amount of Ero-1α increased as the MV4-11 treated with CB5083 at 200nM increased to 500nM, while in Fig 6 the high level of Ero-1α gradually decreased in the CB5083-resistant MV4-11 treated with CB5083 from 200nM to 500nM. Could the authors explain a little bit on this issue?

The main difference between Fig 1C and Fig 6A is the duration of exposure to the VCP inhibitor CB-5083. Expression of Ero-1α is induced in response to ER stress. However, we believe that cells exposed to ER stress for extended time periods gradually adapt their ER stress response likely resulting in reduced levels of Ero-1α.

4, As shown in Fig 6, the CB5083-resistant MV4-11 cells led by VCP gene mutation exhibited permanent ER stress, but no apoptosis occurred. Please briefly discuss the possible mechanism of this phenomena in DISCUSSION, especially on the relationship among UPR, ER stress and apoptosis.

Aim of the unfolded protein response is to restore ER protein homeostasis however prolonged unfolded protein response can lead to induction of apoptosis. CHOP is considered a central regulator of ER stress induced apoptosis and its expression can be induced by the PERK–eIF2α–ATF4 as well as IRE1–TRAF2 signaling axis. Interestingly, we observe reduced levels of PERK and IRE1 in cells that have been exposed to the VCP inhibitor CB-5083 for extended time periods (Fig 1c and Fig 6a). The reduced expression levels of these proteins might be a result of the adaptation to persistent ER stress 

5, The origin of human BM stromal cell line HS-5 should be mentioned in MTERIALS AND METHODS

We added further information about the HS-5 cell line to the Materials and Methods section.

I like to recommend accept this manuscript for publication after the authors revise or explain on these minor points mentioned above.

Reviewer #2: This is an interesting study and the authors have collected a unique dataset using cutting edge methodology. The paper is generally well written and structured. An excellent report on very thorough research. The literature review was thorough; the methodology was painstakingly thorough.

We thank the reviewer for the positive evaluation of our work.

---

## [Editor Report · Decision Letter 1]

22 Mar 2022

VCP inhibition induces an unfolded protein response and apoptosis in human acute myeloid leukemia cells

PONE-D-21-30344R1

Dear Dr. Wagner,

We’re pleased to inform you that your manuscript has been judged scientifically suitable for publication and will be formally accepted for publication once it meets all outstanding technical requirements.

Kind regards,

Anilkumar Gopalakrishnapillai

Academic Editor

PLOS ONE

---

## [Editor Report · Acceptance letter]

24 Mar 2022

PONE-D-21-30344R1 

VCP inhibition induces an unfolded protein response and apoptosis in human acute myeloid leukemia cells 

Dear Dr. Wagner:

I'm pleased to inform you that your manuscript has been deemed suitable for publication in PLOS ONE. Congratulations! Your manuscript is now with our production department. 

Kind regards, 

on behalf of

Dr. Anilkumar Gopalakrishnapillai 

Academic Editor

PLOS ONE